# An Efficient 2D DOA Estimation Algorithm Based on OMP for Rectangular Array

Chuang Wang [2,3,4], Jianmin Hu [1,2,*,†], Qunying Zhang [2,3,4,*,†] and Xinhao Yuan [1]

1   GBA Branch of Aerospace Information Research Institute, Chinese Academy of Sciences, Guangzhou 510530, China
2   Aerospace Information Research Institute, Chinese Academy of Sciences, Beijing 100094, China
3   Key Laboratory of Electromagnetic Radiation and Sensing Technology, Chinese Academy of Sciences, Beijing 100190, China
4   School of Electronic, Electrical and Communication Engineering, University of Chinese Academy of Sciences, Beijing 100049, China
*   Correspondence: hujm@aircas.ac.cn (J.H.); qyzhang@aircas.ac.cn (Q.Z.)
†   These authors contributed equally to this work.

**Abstract:** Recently, orthogonal matching pursuit (OMP) has been widely used in direction of arrival (DOA) studies, which not only greatly improves the resolution of DOA, but can also be applied to single-snapshot and coherent source cases. When applying the OMP algorithm to the rectangular array DOA of the millimeter-wave radar, it is necessary to reshape the two-dimensional (2D) signal into a long one-dimensional (1D) signal. However, the long 1D signal will greatly increase the number and length of atoms in the complete dictionary of the OMP algorithm, which will greatly increase the amount of computation. Taking advantage of the sparsity of targets in the DOA space, an efficient 2D DOA estimation algorithm based on OMP for rectangular array is proposed. The main idea is to reduce the number of atoms in the complete dictionary of the OMP algorithm, thereby greatly reducing the amount of computation required. A simulation verifies that the efficiency of the proposed algorithm is much higher than the conventional algorithm with almost the same estimation accuracy.

**Keywords:** orthogonal matching pursuit (OMP); direction of arrival (DOA); two-dimensional (2D); rectangular array

## 1. Introduction

Recently, autonomous driving has become a fast-growing field of research, which is an important direction to achieve safe and efficient driving. Compared with other sensors, millimeter-wave radar has the advantages of being usable at any time and in all weather, motion detection capability, and low cost, and is one of the most important sensors for the automatic driving of cars [1]. In order to better realize the perception of three-dimensional space, two-dimensional (2D) direction of arrival (DOA) [2–4] estimation is an important area of research in the application of vehicle millimeter-wave radar. L-shaped arrays [5], rectangular arrays [6] and circular arrays [7] are usually used to realize the estimation of 2D DOA. Sufficient array snapshots are required to achieve precise angle estimation due to the low density of both L-shaped and circular arrays. However, in vehicle-mounted millimeter-wave radar application scenarios, only a small number of array snapshots, or even just a single array snapshot can be obtained due to the highly dynamic scene [8,9]. Therefore, in this highly automotive dynamic scenario, L-shaped arrays and circular arrays are not suitable for accurate 2D angle estimation. Rectangular arrays offer advantages over L-shaped and circular arrays in that they not only enable accurate angle estimation with only a small number of array snapshots, but also can be implemented with a few antennas

through multi-input multi-output (MIMO) radar technology [10,11]. Because of the above-mentioned advantages, rectangular arrays have been widely used in vehicle-mounted millimeter-wave radar and various other domains.

In recent years, compressed sensing algorithms have been developed rapidly and widely used in image fusion [12], ultra-wideband communication [13], and underwater acoustic communication [14]. The high bandwidth of the millimeter-wave radar enables it to achieve a high range resolution [15]. Thus, the number of targets in the same range-Doppler bin is limited, indicating the sparsity of targets in the DOA space. According to this characteristic, compressed sensing methods can be used for high-resolution angle estimation of the millimeter-wave radar. The DOA estimation based on compressed sensing [16] not only improves the resolution of DOA greatly, but can also be applied to the case of single-snapshot and coherent sources compared with the classical DOA estimation algorithms, such as MVDR [17], MUSIC [18], ESPRIT [19], etc. Recently, many interesting angle estimation methods have been proposed. Yardibi [20] proposes an iterative adaptive approach (IAA), which can obtain high angle resolution through an iterative process. Tan [21] proposes a sparse learning via iterative minimization method for sparse signal recovery, which is essentially a regularization minimization method. Pallotta [22] proposes a two-level interpolation method to perform angle estimation to improve the accuracy of angle estimation. Ciuonzo [23] derives the asymptotic distribution of the null spectrum of the well-known MUSIC in its computational time-reversal form, and provides useful numerical analysis to validate the theory. Although the accuracy of the above algorithms is high enough, their calculation amount is so huge that they are very hard to apply in the actual scenarios. Recently, there are also many works to improve the efficiency of the DOA algorithm. Mao [24] proposed an efficient generalized adaptive asymptotic minimum variance method to improve the angular resolution. However, it is based on the sparse asymptotic minimum variance algorithm [25], which has difficulty in controlling the end of iteration [26]. Luo [27] proposed an online least absolute shrinkage and selection operator method to achieve efficient DOA estimation. However, it is designed for the time-division multiplexing multiple-input, multiple-output system, which greatly limits its application range. Orthogonal matching pursuit [28,29] (OMP) is one of the classic algorithms in the field of compressed sensing. It is a greedy algorithm with the advantages of simple implementation, stable performance and low computational complexity.

When applying the OMP algorithm to 2D DOA estimation of the rectangular array for the vehicle-mounted millimeter-wave radar, it is necessary to reshape the 2D signal into a one-dimensional (1D) signal. However, the 1D signal is excessively lengthy, leading to a significant increase in the corresponding dictionary's length and number of atoms. As a result, the OMP algorithm's computational complexity is considerably elevated [30,31], which cannot meet the real-time requirements of the vehicle-mounted millimeter-wave radar.

An efficient 2D DOA algorithm based on OMP for the rectangular array is proposed by exploiting the sparsity of the targets. Initially, a 1D digital beamforming (DBF) [32,33] is conducted in each dimension, resulting in the DBF power spectrum. Subsequently, a peak search operation is carried out on the DBF result of each dimension, and the peaks' corresponding intermediate angle set and pitch angle set are returned. Next, these two angle sets are extended by the method mentioned in the manuscript. Finally, a complete dictionary is generated based on the two extended angle sets, and the OMP algorithm is used to reconstruct the sparse vector according to the complete dictionary and the sparse signal. Finally, the angles of targets can be obtained according to the position of the non-zero elements in the sparse vector.

The main contribution of our proposed algorithm is to solve the problem of excessive computation in the OMP-based 2D DOA estimation algorithm. We introduce the 1D DBF, peak search, and angle set extension operation to reduce the number of atoms greatly in the complete dictionary, which greatly reduces the computational complexity of the OMP-based 2D DOA estimation algorithm, and the angle set extension operation introduced

by us is the key to ensuring the estimation accuracy and robustness of our proposed algorithm. Compared with the conventional OMP-based 2D DOA estimation algorithm, our proposed algorithm greatly reduces the amount of computation while maintaining estimation accuracy. Therefore, our proposed algorithm has broad application prospects in scenarios requiring real-time performance.

This paper is organized as follows: Section 2 presents the signal model. Our proposed 2D DOA algorithm based on OMP is introduced in Section 3. In Section 4, we apply our proposed algorithm along with other state-of-the-art techniques to process the simulation data and demonstrate the effectiveness of our approach. Finally, Section 5 provides the conclusions.

## 2. Signal Model of Rectangular Array

Figure 1 illustrates a rectangular array with $M \times N$ array elements distributed on the $x$-$z$ plane. To simplify the modeling process, the array is assumed to be uniform, with adjacent array elements separated by distances of $d_x$ in the $x$-axis direction and $d_z$ in the $z$-axis direction. Assuming that the echo signals of the targets come from the far field, the number of the targets is $K$, the complex reflection coefficient, the azimuth angle and the pitch angle corresponding to the $k$-th target are $\beta_k$, $\theta_k$ and $\varphi_k$, respectively. Assuming no interference from noise, the response with respect to $K$ targets of the $(m, n)$-th array element can be represented as

$$s_{mn} = \sum_{k=1}^{K} \left\{ \beta_k \exp\left(j2\pi \frac{d_x}{\lambda}(m-1)\sin \alpha_k\right) \cdot \exp\left(j2\pi \frac{d_z}{\lambda}(n-1)\sin \varphi_k\right) \right\}, \tag{1}$$

where $\lambda$ is the carrier wavelength, $m$, $n$ are positive integers, and their value ranges are $1 \leq m \leq M$, $1 \leq n \leq N$, and $\alpha_k$ is defined as the intermediate angle variable, which is a combination of the azimuth angle $\theta_k$ and the pitch angle $\varphi_k$. The relationship between them is

$$\sin \alpha_k = \sin \theta_k \cos \varphi_k. \tag{2}$$

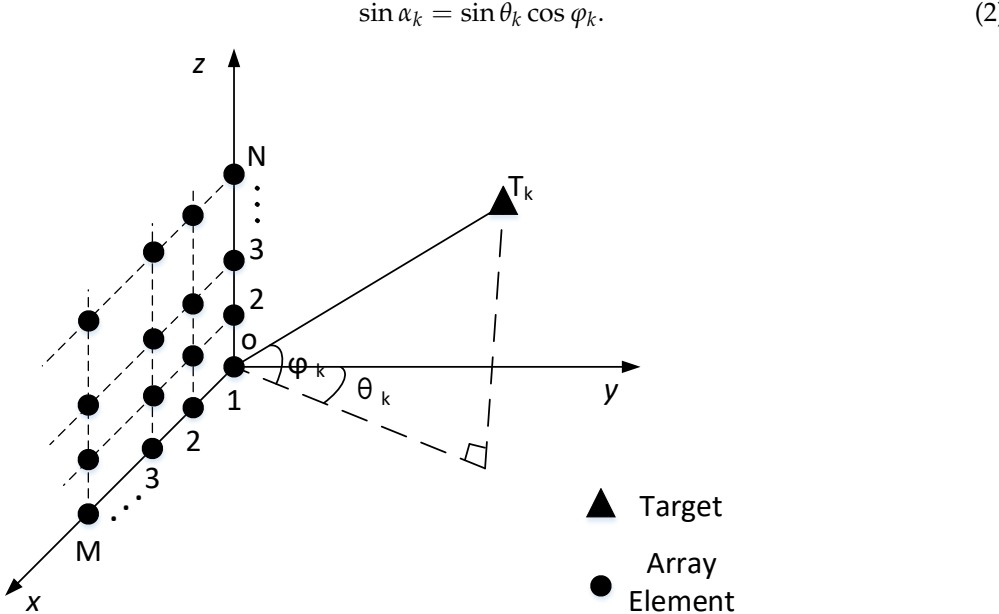

**Figure 1.** Geometry of rectangular array.

The main idea of 2D DOA for the rectangular array is to obtain the intermediate angle $\alpha_k$ and the pitch angle $\varphi_k$ first, and then calculate the azimuth angle $\theta_k$ according to (2). In the later discussion, only the intermediate angle $\alpha_k$ and the pitch angle $\varphi_k$ are considered.

### 3. Proposed 2D DOA Algorithm Based on OMP

*3.1. Perform 1D DBF in Each Dimension*

The responses of the first row array elements and the first column array elements are taken out, and 1D digital beamforming (DBF) is performed on them, respectively. The row is defined as the x-axis direction, while the column is defined as the z-axis direction in this manuscript.

The response of the first row array elements to the targets is

$$S_{row1} = \begin{bmatrix} s_{11} \\ s_{21} \\ s_{31} \\ \dots \\ s_{M1} \end{bmatrix}_{M \times 1}. \tag{3}$$

The steering vector corresponding to the first row of array elements is

$$v_{row1}(\alpha) = \begin{bmatrix} f_{row1}(\alpha, 1) \\ f_{row1}(\alpha, 2) \\ f_{row1}(\alpha, 3) \\ \dots \\ f_{row1}(\alpha, M) \end{bmatrix}_{M \times 1}, \tag{4}$$

where

$$f_{row1}(\alpha, m) = \exp\left( j2\pi \frac{d_x}{\lambda}(m-1)\sin\alpha \right). \tag{5}$$

Perform uniform discrete sampling of the intermediate angle $\alpha$

$$\alpha_p = 2\frac{p - (P-1)/2}{P}\alpha_{\max}, p = 1, 2, 3, \dots, P, \tag{6}$$

where the total number of discrete points $P$ is assumed to be odd, and $\alpha_{\max}$ is defined as the maximum unambiguous intermediate angle, which can be calculated as

$$\alpha_{\max} = \arcsin\left( \frac{\lambda}{2d_x} \right). \tag{7}$$

The set of the discrete intermediate angles can be written as

$$A = \{\alpha_1, \alpha_2, \alpha_3, \dots, \alpha_P\}. \tag{8}$$

The steering matrix based on the discrete intermediate angles is

$$V_{row1} = [v_{row1}(\alpha_1), v_{row1}(\alpha_2), v_{row1}(\alpha_3), \dots, v_{row1}(\alpha_P)]_{M \times P}. \tag{9}$$

One-dimensional DBF in the first row is performed by multiplying the transpose conjugate of (9) and (3)

$$DBF_{row1}(\alpha_p) = V_{row1}^H S_{row1}, p = 1, 2, 3, \dots, P, \tag{10}$$

where $(\cdot)^H$ denotes the transpose conjugate.

The response of the first column of array elements to the targets is

$$S_{col1} = \begin{bmatrix} s_{11} \\ s_{12} \\ s_{13} \\ \dots \\ s_{1N} \end{bmatrix}_{N \times 1}. \tag{11}$$

The steering vector corresponding to the first column of array elements is

$$v_{col1}(\varphi) = \begin{bmatrix} f_{col1}(\varphi,1) \\ f_{col1}(\varphi,2) \\ f_{col1}(\varphi,3) \\ \dots \\ f_{col1}(\varphi,N) \end{bmatrix}_{N\times 1}, \tag{12}$$

where

$$f_{col1}(\varphi,n) = \exp\left(j2\pi\frac{d_z}{\lambda}(n-1)\sin\varphi\right). \tag{13}$$

Perform uniform discrete sampling on the pitch angle $\varphi$

$$\varphi_q = 2\frac{q-(Q-1)/2}{Q}\varphi_{\max}, q = 1,2,3,\dots,Q, \tag{14}$$

where the total number of discrete points $Q$ is assumed to be odd, and $\varphi_{\max}$ is defined as the maximum unambiguous pitch angle, which can be calculated as

$$\varphi_{\max} = \arcsin\left(\frac{\lambda}{2d_z}\right). \tag{15}$$

The set of the discrete pitch angles can be written as

$$\phi = \{\varphi_1,\varphi_2,\varphi_3,\dots,\varphi_Q\}. \tag{16}$$

The steering matrix based on the discrete pitch angles is

$$V_{col1} = [v_{col1}(\varphi_1),v_{col1}(\varphi_2),v_{col1}(\varphi_3),\dots,v_{col1}(\varphi_Q)]_{N\times Q}. \tag{17}$$

One-dimensional DBF in the first column is performed by multiplying the transpose conjugate of (17) and (11)

$$DBF_{col1}(\varphi_q) = V_{col1}^H S_{col1}, q = 1,2,3\dots,Q. \tag{18}$$

### 3.2. Search Peaks on the 1D DBF Results

Peak search operation is performed on the 1D DBF results $DBF_{row1}(\alpha_p)$ and $DBF_{col1}(\varphi_q)$. This operation specifically refers to finding all peaks of the DBF power spectrum, which are greater than a certain discrimination threshold, and the angular coordinates corresponding to these peaks are outputted.

Assuming that $K_1$ and $K_2$ peaks are found for $DBF_{row1}(\alpha_p)$ and $DBF_{col1}(\varphi_q)$, respectively. The intermediate angle set and pitch angle set corresponding to these peaks can be respectively recorded as

$$U = \{u_{k_1} \in A | k_1 = 1,2,3,\dots,K_1\}, \tag{19}$$

$$W = \{w_{k_2} \in \phi | k_2 = 1,2,3,\dots,K_2\}, \tag{20}$$

where $u_{k_1}$ is the azimuth angle corresponding to the $k_1th$ peak in the 1D DBF results in row, and $w_{k_2}$ is the intermediate angle corresponding to the $k_2th$ peak in the 1D DBF results in column.

### 3.3. Extend the Intermediate Angle Set and Pitch Angle Set

When the angular difference between two targets is less than the angular resolution, the two targets cannot be distinguished by using the DBF operation. In this case, the angle set obtained above deviates greatly from the angle set of the actual targets. Therefore, the

angle set obtained in Section 3.2 needs to be extended to ensure that it contains the angle set of the actual targets. The specific angle set extension method is shown as follows:

The new intermediate angle set $D$ can be extended from the angle set $U$ as

$$
\begin{aligned}
D = \{d \in A | u_1 - \alpha_{res} \leq d \leq u_1 + \alpha_{res}, \\
u_2 - \alpha_{res} \leq d \leq u_2 + \alpha_{res}, \\
u_3 - \alpha_{res} \leq d \leq u_3 + \alpha_{res}, \\
\ldots, \\
u_{K_1} - \alpha_{res} \leq d \leq u_{K_1} + \alpha_{res} \},
\end{aligned}
\tag{21}
$$

where $\alpha_{res}$ is the intermediate angle resolution, and it can be calculated as

$$
\alpha_{res} = \frac{\lambda}{M d_x \cos(\alpha)}.
\tag{22}
$$

The new pitch angle set $E$ can be extended from the pitch angle set $W$ as

$$
\begin{aligned}
E = \{e \in \phi | w_1 - \varphi_{res} \leq e \leq w_1 + \varphi_{res}, \\
w_2 - \varphi_{res} \leq e \leq w_2 + \varphi_{res}, \\
w_3 - \varphi_{res} \leq e \leq w_3 + \varphi_{res}, \\
\ldots, \\
w_{K_2} - \varphi_{res} \leq e \leq w_{K_2} + \varphi_{res} \},
\end{aligned}
\tag{23}
$$

where $\varphi_{res}$ is the pitch angle resolution, and it can be calculated as

$$
\varphi_{res} = \frac{\lambda}{N d_z \cos(\varphi)}.
\tag{24}
$$

Suppose there are $K_3$ elements in the set $D$ and $K_4$ elements in the set $E$. At this time, $D$ and $E$ can be simply written as

$$
D = \{d_1, d_2, \ldots, d_{K_3}\},
\tag{25}
$$

$$
E = \{e_1, e_2, \ldots, e_{K_4}\}.
\tag{26}
$$

### 3.4. OMP Algorithm

In order to apply the OMP algorithm to rectangular array DOA, the 2D response needs to be reshaped into a 1D response as follows:

$$
\begin{aligned}
S_{all} = [s_{11}, s_{21}, \ldots, s_{M1}, \\
s_{12}, s_{22}, \ldots, s_{M2}, \\
s_{13}, s_{23}, \ldots, s_{M3}, \\
\ldots, \\
s_{1N}, s_{2N}, \ldots, s_{MN}]_{1 \times MN}^{T},
\end{aligned}
\tag{27}
$$

where $(\cdot)^{T}$ denotes the transpose.

The steering vector of the array corresponding to the 1D response in (27) is

$$
v_{all}(\alpha, \varphi) = [kron(v_{col1}(\varphi), v_{row1}(\alpha))]_{MN \times 1},
\tag{28}
$$

where $kron(\cdot)$ stands for the Kronecker product [34].

Then, the steering matrix of this array under the angle set $D$ and $E$ is

$$
\begin{aligned}
V_{all} = \big[ & v_{all}(d_1,e_1), v_{all}(d_2,e_1), \ldots, v_{all}(d_{K_3},e_1), \\
& v_{all}(d_1,e_2), v_{all}(d_2,e_2), \ldots, v_{all}(d_{K_3},e_2), \\
& v_{all}(d_1,e_3), v_{all}(d_2,e_3), \ldots, v_{all}(d_{K_3},e_3), \\
& \qquad\qquad \ldots, \\
& v_{all}(d_1,e_{K_4}), v_{all}(d_2,e_{K_4}), \ldots, v_{all}(d_{K_3},e_{K_4}) \big]_{MN \times K_3 K_4}.
\end{aligned}
\tag{29}
$$

The steering matrix $V_{all}$ can be regarded as the complete redundant dictionary, and $S_{all}$ in (27) can be regarded as the observation signal. The following sparse linear model can be constructed

$$
S_{all} = V_{all} x, \tag{30}
$$

where $x$ is the corresponding sparse vector. The sparsity of $x$ is $K$, and the size of $x$ is $K_3 K_4 \times 1$.

The sparse vector $x$ in (30) can be reconstructed by the OMP algorithm. Finally, the angles of targets can be obtained by finding positions of non-zero elements in $x$.

The processing flow of the algorithm is summarized as shown in Figure 2. First, perform a 1D DBF in each dimension. Then, perform a peak search operation in the DBF result of each dimension, and return the set of intermediate angles and pitch angles corresponding to the peaks found above. Next, extend these two angle sets using the method mentioned in this manuscript. Finally, a complete dictionary is generated based on the two extended angle sets, the OMP algorithm is used to reconstruct the sparse vector according to the complete dictionary and the sparse signal, and the angles of targets can be obtained according to the positions of the non-zero elements in the sparse vector.

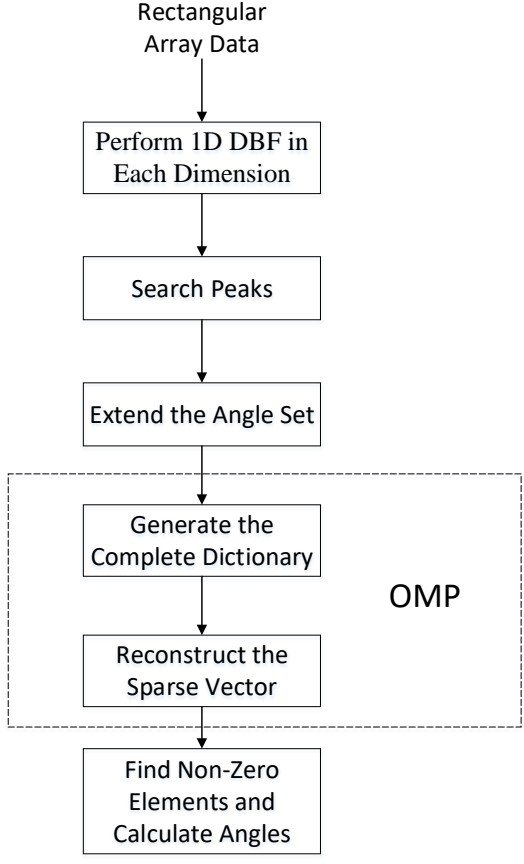

**Figure 2.** Flow chart of our proposed algorithm.

### 3.5. Comparison of Computational Complexity

The conventional 2D DOA based on OMP has $PQ$ atoms, whereas our proposed algorithm has $K_3K_4$ atoms. In OMP, the most time-consuming step is to search atoms in the complete dictionary. In each iteration, the conventional method searches atoms at a cost of O($MNPQ$), resulting in a total complexity of O($KMNPQ$). Conversely, our method searches atoms at a cost of only O($MNK_3K_4$) in each iteration, giving it a total complexity of O($KMNK_3K_4$). As the targets are sparse, $K_3K_4$ is much smaller than $PQ$, so our algorithm's computational complexity is much lower than that of the conventional 2D DOA based on OMP.

## 4. Simulation Results

The simulation is carried out in MATLAB R2021a on a PC with Intel® Core™ i7-9700 CPU @3.00 GHz and 16 GB RAM, and the system is 64-bit operating system-Microsoft Windows 11.

To mitigate the impact of randomness on algorithm performance assessment, multiple independent random Monte Carlo simulations are conducted to evaluate both the root mean square error (RMSE) and the average time consumption. The RMSE is defined as

$$
\begin{aligned}
&RMSE \\
&= \sqrt{\frac{1}{N_{mc}K} \sum_{n=1}^{N_{mc}} \sum_{k=1}^{K} \left[ \left( \hat{\alpha}_{n,k} - \alpha_{n,k} \right)^2 + \left( \hat{\varphi}_{n,k} - \varphi_{n,k} \right)^2 \right]},
\end{aligned}
\tag{31}
$$

where $N_{mc}$ is the number of Monte Carlo simulations. Variables $\hat{\alpha}_{n,k}$ and $\alpha_{n,k}$ are the estimated value and the true value of the intermediate angle of the $k$-th target in the $n$-th independent simulation, respectively. Variables $\hat{\varphi}_{n,k}$ and $\varphi_{n,k}$ are the estimated value and the true value of the pitch angle of the $k$-th target in the $n$-th independent simulation, respectively.

Table 1 lists the simulation parameters, and there are $K = 3$ far-field targets in the simulation. For a given signal-to-noise ratio (SNR), $N_{mc} = 1500$ trials Monte Carlo simulations are made. In each trial, the angles of these three targets are randomly generated in the range of $[-50°, 50°]$. Simulation results are shown in Figure 3 and Table 2. Figure 3 shows the average RMSE variation of the estimated angles of all three targets in 1500-trials Monte Carlo simulations, where the OMP-based 2D DOA, IAA-based 2D DOA, DBF-based 2D DOA, and our proposed algorithm are compared. Table 2 shows the average time consumed by the above four algorithms in 1500-trials Monte Carlo simulations. It can be inferred that our proposed algorithm is capable of achieving a similar estimation performance to the conventional 2D DOA method based on OMP. Moreover, our proposed algorithm consumes significantly less time on average compared to the conventional algorithm. The RMSE of IAA-based 2D DOA is lower than that of OMP-based 2D DOA, DBF-based 2D DOA, and our proposed algorithm, which means that the estimation accuracy of IAA-based 2D DOA is highest. However, its computational load is very large, which makes it difficult to apply in vehicular radar. Since DBF-based 2D DOA is not a super-resolution algorithm, its accuracy is the worst among these four DOA algorithms, and it cannot meet the requirements of some application scenarios that require high angular resolution.

When the two targets are too close, the angle set obtained by the 1D DBF operation will be inaccurate. Therefore, we design the angle set extension operation to enhance the robustness of our proposed algorithm. In order to verify the effectiveness of the angle set extension operation, we have designed the following simulation.

When SNR = 15 dB, the 2D angles of two targets are set as $(9°, 20°)$ and $(8°, 40°)$, respectively. The simulation results are shown in Table 3. It can be seen that our proposed algorithm can distinguish the two targets in both dimensions, while our proposed algorithm (without the angle set extension operation) can only distinguish the two tar-

gets in the second dimension. The simulation results prove the effectiveness of the angle extension operation, which is the key step to ensure the accuracy and robustness of our proposed algorithm.

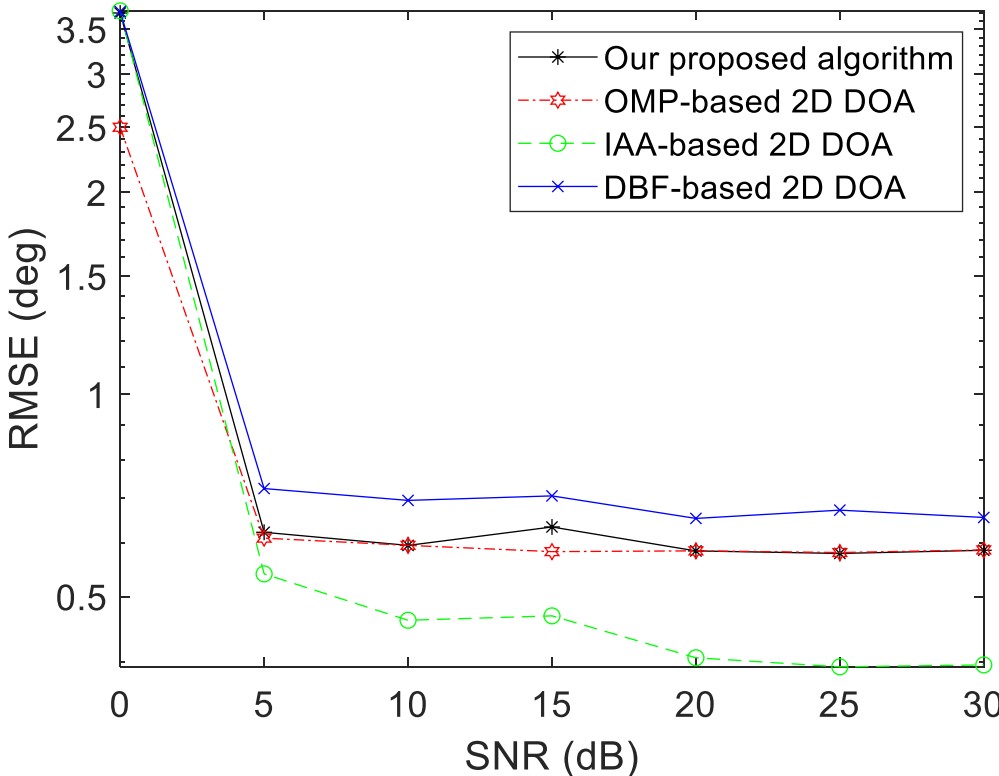

**Figure 3.** Average RMSE of the estimated angles of three targets in 1500-trials Monte Carlo simulations.

**Table 1.** Main simulation parameters.

| Parameter Description | Symbol | Value | Unit |
|---|---|---|---|
| Light Speed | $c$ | $3 \times 10^8$ | m/s |
| Central Frequency | $f_c$ | $77 \times 10^9$ | Hz |
| Wavelength | $\lambda$ | $c/f_c$ | m |
| Snapshot Number | | 1 | |
| Target Number | $K$ | 3 | |
| Element Number per Row | $M$ | 20 | |
| Element Number per Column | $N$ | 20 | |
| Adjacent Element Distance in Row | $d_x$ | $\lambda/2$ | m |
| Adjacent Element Distance in Column | $d_z$ | $\lambda/2$ | m |
| Randomized Simulation Number | $N_{mc}$ | 1500 | |
| Discrete Number of Intermediate Angle | $P$ | 181 | |
| Discrete Number of Pitch Angle | $Q$ | 181 | |

**Table 2.** The average time consumed in 1500-trials Monte Carlo simulations.

| Algorithms | Average Time Consumed (s) |
|---|---|
| OMP-based 2D DOA | 0.3934 |
| DBF-based 2D DOA | 0.0286 |
| Our proposed algorithm | 0.0244 |
| IAA-based 2D DOA | 15.1074 |

**Table 3.** DOA estimation results.

|  | Target 1 | Target 2 |
|---|---|---|
| 2D angle of target | $(9°, 20°)$ | $(8°, 40°)$ |
| OMP-based 2D DOA | $(9°, 20°)$ | $(8°, 40°)$ |
| IAA-based 2D DOA | $(9°, 20°)$ | $(8°, 40°)$ |
| DBF-based 2D DOA | $(9°, 20°)$ | $(8°, 40°)$ |
| Our proposed algorithm | $(9°, 20°)$ | $(8°, 40°)$ |
| Our proposed algorithm (without the angle set extension operation) | $(9°, 20°)$ | $(9°, 40°)$ |

## 5. Conclusions

Taking advantage of the sparsity of the targets in the same range-Doppler bin of the millimeter-wave radar, an efficient 2D DOA estimation algorithm based on OMP for the rectangular array is proposed. Compared with the conventional OMP-based 2D DOA estimation algorithm, our proposed algorithm can greatly reduce the number of atoms in the complete dictionary, thereby greatly reducing the amount of calculation required and greatly improving the estimation efficiency under the premise of ensuring the estimation accuracy. This has important application prospects in some scenarios requiring real-time high-precision 2D DOA, such as in the application of vehicle-mounted millimeter wave radar.

This paper mainly discusses how to improve the efficiency of the OMP algorithm so that it can be applied to 2D DOA estimation of the automotive radar. Although subspace-based DOA estimation methods have high resolutions, they need the accumulation of multiple snapshots, which affects the real-time performance of them. In the future, it will also be meaningful to apply subspace-based DOA methods, such as MUSIC and ESPRIT, to automotive radar.

**Author Contributions:** Conceptualization, C.W.; Data curation, X.Y.; Formal analysis, C.W.; Funding acquisition, J.H. and Q.Z.; Investigation, C.W.; Project administration, J.H. and Q.Z.; Resources, X.Y.; Software, C.W.; Supervision, J.H., Q.Z. and X.Y.; Validation, C.W.; Writing—original draft, C.W.; Writing—review and editing, C.W. All authors have read and agreed to the published version of the manuscript.

**Funding:** This work was supported in part by the National Natural Science Foundation of China under Grant 61988102; in part by the initial funding of GBA branch of Aerospace Information Research Institute, Chinese Academy of Sciences under Grant 2019B090909011; in part by the Key-Area Research and Development Program of Guangdong Province under Grant 2019B090917007; in part by the National Key Research and Development Program of China under Grant 2017YFA0701004, and Grant 2020YFC1522202-4; in part by the Project of Equipment Pre-Research under Grant WJ2019G00019; in part by the Pearl River Talent Plan under Grant 2021QN02Z134.

**Conflicts of Interest:** The authors declare no conflict of interest.

## Abbreviations

The following abbreviations are used in this manuscript:

| | |
|---|---|
| OMP | Orthogonal matching pursuit |
| DOA | Direction of arrival |
| 2D | Two-dimensional |
| 1D | One-dimensional |
| MIMO | Multi-input multi-output |
| DBF | Digital beamforming |
| RMSE | Root mean square error |
| SNR | Signal to noise ratio |
| IAA | Iterative adaptive approach |

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
