# Peer review of "An Efficient 2D DOA Estimation Algorithm Based on OMP for Rectangular Array"

_electronics, doi:10.3390/electronics12071634_

Round 1

Reviewer 1 Report

Main comments

This paper proposes an orthogonal matching pursuit based two-dimensional direction of arrival estimation algorithm that has lower computational complexity than that of the reference algorithm while maintaining similar accuracy. Thus, the main (yet limited) novelty is in improving the computational efficiency (complexity reduction). The overall structure of the paper is decent and related literature is sufficiently covered. There are some minor issues to be addressed to improve the manuscript as listed below under ‘Minor comments’.

Minor comments

1.       Abstract, page 1, line numbers 1-2: The expression ‘in the field of direction of arrival (DOA)’ requires some additional text at the end, e.g., ‘in the field of direction of arrival (DOA) research’ or ‘in the field of direction of arrival (DOA) studies’ to be complete.

2.       Keywords, page 1, line number 13: array -> array.

3.       Introduction, page 1, line number 17: communication[1] -> communication [1]                               Similarly, in the rest of the manuscript, separate reference numbering from preceding text by a single space.

4.       Introduction, page 1, line number 27: highly automotive dynamic scene -> highly dynamic automotive scene (scene -> scenario?)

5.       Introduction, page 1, line numbers 31-32: Because of above-mentioned advantages -> Because of the above-mentioned advantages

6.       Section 2.1, page 2, (2): As the equation closes the sentence there should be a full-stop at the end of it. Check and correct the rest of the manuscript in this regard where needed.

7.       Section 2.1, page 3, Figure 1: Put a full-stop at the end of figure caption text. The same applies to other figures in the manuscript.

8.       Section 2.1, page 3, line number 80: Equation(2) -> (2)

       In the middle of the sentence equations are presented simply as        numbers in parentheses. If it begins sentence, then, e.g., Equation (2) …

9.       Section 2.2.1, page 3, line number 86: them respectively -> them, respectively

10.   Section 2.2.2, page 5, line number 115: ) respectively -> ), respectively

11.   Section 2.2.3, page 5, line numbers 125-126: Section 2.2.2 -> Section 2.2.2 as

12.   Section 2.2.3, page 5, line numbers 128-129: Section 2.2.2 -> Section 2.2.2 to be

13.   Section 2.2.4, page 6, line number 146-148: Please, avoid short one sentence paragraphs. (In most cases, a couple of consecutive sentences form a paragraph.)

14.   Section 3, page 8, Table 1 caption: main simulation parameters -> Main simulation parameters.

15.   Section 3, page 8, line numbers 175 and 177: Avoid beginning a sentence with a small letter or a small-lettered symbol. You can put ahead the word ‘Symbol’, ‘Notation’, ‘Variable’ or something like that.

16.   Section 3, page 8, line number 180: trails -> trials

The same applies to the rest of the manuscript, e.g., lines 181, 184, and 186.

17.   Section 3, page 9, Figure 3: In x- and y-axis labels it would be better to avoid symbol / as a separator for metric and unit. For example, in x-axis SNR (dB) or SNR [dB] would be more widely accepted notation.

Reviewer 2 Report

This paper needs more novelty. The reviewer needs help finding whether the idea proposed in this paper is better than those in other papers. Please point out the paper's originality in the "Introduction" section.

The author should pay more attention to the structure of the article. In Section "Introduction," the reviewer cannot find Sect's writing objective and contribution in the reviewer's opinion, Sect. "Introduction" should point out clearly which problem is studied in this paper, what are the shortages of the existing methods, how to solve the problem by the authors, and what the advantages of the approach proposed by the authors. Please rewrite this section.

Authors must compare with existing state-of-the-art techniques to validate the proposed solution.

Why are the results almost equal after the 5 dB of SNR? 

Table 2 shows the proposed algorithm's average time consumption is much less than the existing one. What is the main reason for that? Are both algorithms have the same simulation parameters?  

Reviewer 3 Report

This paper deals with the problem of 2D DOA estimation exploiting sparse reconstruction. In particular, the problem is firstly reduced to two 1D DOA estimation problems and then the final 2D solution is found. The grand advantage of this method is its low computational complexity with respect it 2D standard counterpart. The paper is in general well-written and interesting, but before publication it needs some improvements.

Firstly, in the review of Literature in Introduction some other interesting papers should be discussed, e.g.:

[1] "Sparse learning via iterative minimization with application to MIMO radar imaging." IEEE TSP 2010.

[2] "DOA refinement through complex parabolic interpolation of a sparse recovered signal." IEEE SPL 2021.

[3] "Noncolocated time-reversal MUSIC: High-SNR distribution of null spectrum." IEEE SPL 2017.

The analyses section should be improved. I suggest adding other scenarios of comparison, for instance the authors could consider both the cases of 1, 2 and 3 targets. Moreover, the RMSE could be represented in a logarithmic scale to better emphasize the differences between curves.

The CRLB should be also include to show the performance benchmark for the estimation algorithms.

At the end of Conclusions some hints for future works should be provided.

Round 2

Reviewer 2 Report

There is a need to do more simulations, as from the simulation result, it needs to be evident that your proposed algorithm is performing better than the existing one. 

Secondly, the simulation parameters for the comparison should be identical. 

From figure 3, the authors are requested to compare the results with the 1D DOA other than the 2D DOA. Also, the simulation results for the proposed one and 2D DOA are approximately the same. What is the novelty of your proposed algorithm?

Reviewer 3 Report

The authors have satisfactorily replied to my previous concerns and have improved the manuscript. In my opinion, it can be now accepted for publication.

Author Response

Dear Reviewer: 
Thank you very much for your comments concerning our manuscript. Those comments are all valuable and very helpful for revising and improving our paper, as well as the important guiding significance to our researches. 
Thank you again for your time and assistance. 

Round 3

Reviewer 2 Report

I have recommended doing more simulations; there is only one simulation figure and one figure of merit. 

Why are the authors performing fewer simulations? What is the central figure of merit in 1D-DOA? Please elaborate. 

There are many research articles available on 1D-DOA. Why is there a comparison between 1D-DOA and 2-D-DOA? Why are the authors not comparing their findings with 1D-DOA? 

The simulation parameters in Table 1 (authors' response) are for 1D-DOA or 2D-DOA? 

Point 3: From figure 3, the authors are requested to compare the results with the 1D DOA other than the 2D DOA. Please compare the simulation results with 1D-DOA. 

Since the targets are sparse, K3K4 is much smaller than PQ. Please elaborate.
